# High Temperature Alters Anthocyanin Concentration and Composition in Grape Berries of Malbec, Merlot, and Pinot Noir in a Cultivar-Dependent Manner

**DOI:** 10.3390/plants11070926

**Published:** 2022-03-30

**Authors:** Inés de Rosas, Leonor Deis, Yésica Baldo, Juan B. Cavagnaro, Pablo F. Cavagnaro

**Affiliations:** 1Plant Physiology Laboratory, Faculty of Agricultural Sciences, National University of Cuyo, Almirante Brown 500, Mendoza M5528 AHB, Argentina; iderosas@fca.uncu.edu.ar (I.d.R.); ldeis@fca.uncu.edu.ar (L.D.); bcavagnaro@fca.uncu.edu.ar (J.B.C.); 2Plant Physiology Laboratory, Institution of Agricultural Biology of Mendoza, Faculty of Agricultural Sciences, National University of Cuyo and Conicet, Almirante Brown 500, Chacras de Coria M5505 AHB, Argentina; 3National Viticulture Institute (INV), Av. San Martín 430, Mendoza M5528 AHB, Argentina; yesicabaldo@inv.gov.ar; 4National Scientific and Technical Research Council (CONICET), Faculty of Agricultural Sciences, National Agricultural Technology Institute (INTA) E.E.A. La Consulta, National University of Cuyo and Conicet, Ex Ruta 40 s/n, San Carlos, La Consulta 5567, Mendoza M5528 AHB, Argentina

**Keywords:** grapevine, climate change, high temperatures, anthocyanins, Malbec, Merlot, Pinot Noir

## Abstract

Climate is determinant for grapevine geographical distribution, berry attributes, and wine quality. Due to climate change, a 2–4 °C increase in mean diurnal temperature is predicted by the end of the century for the most important Argentine viticulture region. We hypothesize that such temperature increase will affect color intensity and other quality attributes of red grapes and wines. The present study investigated the effect of high temperature (HT) on anthocyanin concentration and composition, pH, and resveratrol and solids content in berries of three major wine-producing varieties during fruit ripening in two seasons. To this end, a structure that increased mean diurnal temperature by 1.5–2.0 °C at berry sites, compared to Control (C) plants grown without such structure, was implemented in field grown vineyards of Malbec, Merlot, and Pinot Noir. Results revealed a cultivar-dependent response to HT conditions, with Malbec and Pinot Noir berries exhibiting significant decreases in total anthocyanin concentration (TAC) at veraison and harvest, respectively, while Merlot maintained an unaffected pigment content under HT. The decrease in TAC was associated with reduced levels of delphinidin, cyanidin, petunidin, peonidin, and malvidin glycosides, and increased ratios of acylated (AA)/non-acylated anthocyanins (NAA), suggesting pigment acylation as a possible stress-response mechanism for attenuating HT negative effects. Under HT, Pinot Noir, which does not produce AA, was the only cultivar with lower TAC at harvest (*p* < 0.05). pH, resveratrol, and solids content were not affected by HT. Our results predict high, medium, and low plasticity with regard to color quality attributes for Malbec, Merlot, and Pinot Noir, respectively, in the context of climate change.

## 1. Introduction

Climate is a determinant for grapevine geographical distribution, berry attributes, and wine quality around the world [1]. Grapevine cultivation in different biogeographical regions demonstrates the great plasticity of this crop plant [1]. However, the effect of climate variations, and especially thermal fluctuation, on phenological behavior and berry quality places vines as a highly sensitive crop and bio-indicators of global warming [2,3,4]. Considering the impact of climate change on viticulture and wine quality, the different adaptability of grapevine cultivars to such new conditions could be the basis for improving resilience of local agricultural systems, in addition to implementing cultivar-specific agricultural practices of the vineyard [5], better if with extensive types in order to preserve the characteristics of the agroecosystem and environmental parameters [6].

The climate in western central Argentina allows the production of high-quality grapes and wines [7]. Argentina is one of the most important grape and wine producing countries, with a grapevine cultivated surface of 214.798 ha and 70.5% of its production concentrated in Mendoza province [8]. The climatological characteristics of this region are optimal for producing high quality grapes and wines [9]. Malbec, Merlot, and Pinot Noir wines from Mendoza are internationally recognized for their high-quality organoleptic characteristics [10]. Malbec is the Argentine emblematic variety, and it is mainly cultivated in Mendoza. Approximately 73.5% of its world production is from Argentina, followed by France (12.5%) and Chile (4%). Merlot is mostly cultivated in France (41.8% of the total production), followed by Italy (8.8%) and USA (7.8%). The cultivated areas with Pinot Noir display a wider geographic distribution. Almost 60% of its production extends over France (27.3%), USA (21.1%), and Germany (9.9%) [11]. Among their high-quality organoleptic characteristics, color intensity, hue, and stability are the main color quality traits in red wines. These attributes are strongly conditioned by the presence, concentration, and composition of anthocyanins in the grape berry skin, along with other constituents that influence color development and stability in grapes and wines (e.g., other non-anthocyanin phenolic compounds) [12].

Anthocyanins accumulate in red grape skin during ripening. Environmental factors such as temperature, exposure to light, and water availability can influence the concentration and composition of anthocyanins [13,14,15,16], the main antioxidant constituents in red grapes. Antioxidant activity of anthocyanins is provided by their flavylium skeleton, which involves radical electron delocalization on sp2 orbitals of the oxonium moiety [17,18]. In addition, the oxidation of the hydroxyl groups positioned at the phenolic groups contributes to their antioxidant capacity as well [17,18]. Particularly, high temperatures were found to reduce berry color and induce changes in the berry chemical composition and physiology in some cultivars [19,20,21,22]. In addition to reduced color, higher proportions of acylated anthocyanins (AA), which are chemically more stable than non-acylated anthocyanins (NAA), were observed in berries of Cabernet Sauvignon and Merlot exposed to high temperatures [16,21]. Additionally, high temperature during ripening generally leads to grapes with greater sugar levels and lower organic acid concentration [23]. In addition, high temperatures during fruit ripening were associated with lower resveratrol content in red wines produced from the cultivars Barbera and Croatina [24]; and other environmental factors and agricultural and technological practices have also been reported to influence the content of this phytochemical in grapes and wines (reviewed by [25]). Furthermore, an inverse association was found between anthocyanin and resveratrol contents, suggesting a counterbalance in their concentrations due the competence for common biosynthetic precursors in these two pathways [26]. Thus, the effect of temperature on anthocyanin levels could, hypothetically, affect resveratrol content in the opposite direction.

Assuming that key environmental factors for grape berry and wine quality—such as temperature—are predicted to vary in the next decades due to climate change [27], it is important to examine the natural plasticity of different cultivars to the predicted thermal increase. Phenotypic plasticity (or adaptability), defined as the ability of an organism (wild and cultivated) to change its phenotype in response to different environments, can help mitigate negative effects due to environmental perturbations [28,29]. For Mendoza region, a 2–4 °C increase in the mean diurnal temperature during summer has been predicted for the end of this century [30,31]. The current data underline the genotype-specific response of grapevines to climate change [1,32]. Climate variations could negatively affect grape and wine quality by impairing color production. Characterizing the effect of high temperatures on anthocyanin content and composition in the main red wine-producing grapes would help estimate their phenotypic plasticity and, perhaps, predict whether they will be able to sustain the high quality wine attributes in the context of climate change.

This study investigated the effect of increased temperatures (∆1.5–2 °C) on anthocyanin content and composition, pH, and resveratrol and solids content in grape berries of Malbec, Merlot, and Pinot Noir, three of the most important Argentine red-wine producing cultivars. The presented data will contribute to adequate regions for growing different grape genotypes under the predicted climate modifications [9] and unveil variations among these major wine-producing cultivars for phenotypic plasticity with regard to their wine quality under such climate change scenario. This information could be useful for mitigating climate change effects employing viticultural practices for resilient and sustainable production.

## 2. Results

### 2.1. Temperature Treatments, Berry Soluble Solids and pH

During grape ripening, mean diurnal temperature was 1.5–2 °C higher in HT than in the C treatment (Figure 1). Maximum temperatures in 2017 for HT and C treatments were 42.5 and 40.5 °C, respectively (∆ = 2 °C); and in 2017, 43.5 and 40.5 °C (∆ = 3 °C). Degree-days (DDs) during the ripening period were higher for the HT treatment in both years (Table 1). No statistical differences were found for berry soluble solids content and pH between the HT and C treatments at all the phenological stages analyzed, for the three cultivars in both years (Table 1).

### 2.2. Anthocyanin Concentration and Composition

HPLC analysis allowed the identification and quantitation of nine anthocyanin pigments in berry skins of Malbec, Pinot Noir, and Merlot (Table 2 and Appendix A). These correspond to five anthocyanin monoglucosides (malvidin glucoside, peonidin glucoside, petunidin glucoside, delphinidin glucoside, and cyanidin glucoside) and four of their acylated derivatives, namely the acetylated and coumarylated forms of malvidin and peonidin glucosides. The latter acylated anthocyanins were detected in Malbec and Merlot, whereas no acylated anthocyanins were detected in Pinot Noir, as expected based on previous reports describing anthocyanin profiles in this cultivar [26,27].

Concentrations of individual anthocyanin pigments, expressed in absolute values (i.e., on a fresh weight basis), for the three grape cultivars grown under HT and C conditions are presented in Appendix A. Malvidin-3-O-glucoside (Mv) was the predominant anthocyanin in all the cultivars, in both years. At harvest, the concentration of this pigment in Control berries of Malbec, Merlot, and Pinot Noir, was 2108.6–2002.9 µg/g^−1^ FW (accounting for 55.3–55.6% of the total anthocyanins content), 1413.8–1295.7 µg/g^−1^ FW (55.9–58.5%), and 1662.6–1136.4 µg/g^−1^ FW (77.6–64.3%), respectively.

Total anthocyanin concentration in Malbec and Pinot Noir berries grown under increased temperature (HT) was significantly lower than their respective berries grown under natural conditions (C), whereas Merlot berries had comparable (not statistically different) anthocyanin contents in both treatments and years (Figure 2). In Malbec berries, these statistical differences between HT and C were consistently observed at veraison in both years, 2017 (*p* < 0.05) and 2018 (*p* < 0.01) (Figure 2A,B); whereas in Pinot Noir significant variation was consistently found at harvest time (but not in earlier phenological states) in 2017 (*p* < 0.01) and 2018 (*p* < 0.01) (Figure 2C,D). In Malbec, the decrease in total anthocyanin content due to high temperatures was 53.8% and 43.6% (relative to concentrations in the Control treatments) in 2017 and 2018, respectively; whereas in Pinot Noir, 13.3% and 16.4% reductions in pigment content were found in 2017 and 2018, respectively.

Relative concentrations (%) of individual anthocyanin pigments in berries of the three grape cultivars are presented in Table 3. The significant reduction in total pigment content observed under HT for Malbec berries at veraison was mainly due to a decrease in concentration of the individual anthocyanins delphinidin (Df) (exhibiting in 2017 and 2018 a percentual decrease of 50–38.5%, respectively), cyanidin (Cn) (50% decrease in both years), petunidin (Pt) (56.3–35% reduction), peonidin (Po) (57.2–37.5% reduction), malvidin (Mv) (53.5–31.9% reduction), and the coumarylated peonidin glycoside (PoCu) (60–33.3% reduction). In Pinot Noir berries under HT, the significant decrease in pigment content found at harvest was caused by a decrease in Mv (15.1–16.7% reduction).

The relative contents (%) of non-acylated and acylated anthocyanins were significantly modified in berries of Malbec and Merlot grown under increased temperature in 2018, but not in 2017 (Figure 3 and Table 3). In 2018, under HT conditions, berries of Merlot significantly increased their relative content of total AA (and consequently decreased their proportion of total NAA) at all three phenological stages, whereas berries of Malbec exhibited similar increases in total AA at half-ripeness and harvest, but not at veraison. In Malbec, the percentual increase in acylated anthocyanins by HT was 6.1% (increasing from 27.7% to 33.8%) at half ripeness, and 2.9% (from 29.8% to 32.7%) at harvest (Figure 3B). This relative increase in acylated anthocyanins was mainly due to a decrease in the relative content of the non-acylated pigments Mv and Po, along with a concomitant increase in their acylated counterparts MvCu, PoCu, and PoAc (Table 3).

For the same year (2018), in Merlot berries, the percentual increase in acylated anthocyanins due to HT was 1.8% at veraison (increased from 8.7% to 10.5%), 2.6% at half ripeness (increased from 11.4% to 14.0%), and 1.6% (increased from 13.0% to 14.6%) at harvest (Figure 3D, and Table 3). This shift in pigment composition is mainly attributable to the acylation of delphinidin, cyanidin, petunidin, and peonidin; as suggested by the significant decrease in relative content of these non-acylated pigments in the HT treatment and the concomitant increase in their acylated counterparts (Table 3).

In 2017, the relative content of total AA also increased (and total NAA decreased) due to HT in Malbec and Merlot in all but one of the six ‘cultivar x phenological stage’ combinations analyzed, but these increases were not statistically significant (Figure 3A,C; Table 3). Despite the absence of significant variation in ‘total AA/total NAA’ ratios due to HT, some individual pigments, both acylated and non-acylated, were significantly affected by the increased temperatures (Table 3). In Pinot Noir berries, which lack acylated anthocyanins, no clear pattern of modifications in the relative composition of non-acylated pigments was observed under high temperature conditions (Table 3). Further information on the relative content of individual anthocyanin pigments for the three grape cultivars under HT and C conditions in both years are presented in Table 3 and Appendix A.

## 3. Discussion

### 3.1. Temperature Regimes

The three grapevine varieties studied in this work were selected because they are major wine-producing varieties worldwide, and because they differ in anthocyanin composition and agronomic performance. Malbec is the most widely grown cultivar in Argentina, and its wine generally exhibits intense and uniform color in most of the country regions where it is cultivated [33]. In contrast, color intensity and the phenolic composition of Merlot wines are strongly influenced by environmental factors, including the growing location [34]. Pinot Noir is the only cultivar—among these three—that does not produce acylated anthocyanins [35,36].

In red wines, color intensity is one of the most important quality traits. Previous studies have shown that the temperature during fruit ripening can influence berry color intensity and quality in Cabernet Sauvignon [21,37], Tempranillo [38], Sangiovese fruiting cuttings [39], and Merlot [40]. In the present work we investigated the effect of temperature increase on anthocyanin content and composition in berries of three major red grape cultivars, using realistic experimental temperature treatments under field conditions in one of the main viticulture regions worldwide. The increase in mean diurnal temperature for Mendoza has been 1.5–2 °C from the past 50 years to present [41], and it is predicted that this figure will increase 2–4 additional degrees by the end of the century [30,31]. Thus, this work anticipates the mean diurnal increment for a half century, according to the regional tendency. 

The possibility that the observed variations in anthocyanin composition may be due to differences in soil composition and/or its temperature is almost negligible, considering that the three grape cultivars were grown in the same location and vineyard, in a quite homogeneous soil, thereby sharing highly similar underground environmental conditions. Additionally, it is unlikely that the heating system applied in the HT treatment (Appendix A) could have induced temperature differences in the volume of soil explored by the root system of the plants, as compared to the Control treatment, considering that: (i) the nylon system used only reached up to the internal edge of the furrow; (ii) under furrow irrigation most of the vine roots tend to grow—and are located—towards the sides of the vine rows, as observed in trial pits in this same vineyard (data not presented); and (iii) in a previous trial using the same system, soil temperature at ~40 cm deep—where most of the roots are—was monitored and no significant differences were found between HT and Control treatments.

### 3.2. Anthocyanin Concentration and Composition in Berries under High Temperature

Total anthocyanin content was significantly reduced in berries of Malbec (~45%) and Pinot Noir (~15%) plants grown under high temperature conditions, whereas such decrease in pigment content was not observed in Merlot. The observed reduction in total pigment content in Malbec was mainly due to a decrease in the concentration (in absolute values) of the individual anthocyanins Df, Cn, Pt, Po, Mv, and PoAc; whereas in Pinot Noir, it was caused by a decrease in Mv (Appendix A). These results suggest that non-acylated anthocyanins are particularly susceptible to degradation by high temperatures. Thus, cultivars with a high proportion of these pigments are less likely to maintain high color intensity—and thereby red wine quality—under a climate change context. Conversely, cultivars with high levels of more stable acylated pigments are expected to be more resilient under such adverse conditions for grape and wine color quality attributes.

Previous studies have reported detrimental effects of high temperatures on anthocyanin content in different grape cultivars (Darkridge, Cabernet Sauvignon, Sangiovese, Tempranillo, Merlot) at biochemical and molecular levels [20,37,38,39,40]. Results from these studies suggest that high temperatures increase anthocyanin degradation, possibly mediated by the action of peroxidase enzymes. Recently, a grapevine peroxidase gene, namely VviPrx31, has been described as a major candidate for HT-mediated anthocyanin degradation in Sangiovese [42]. In addition to accelerating pigment degradation, it has been reported that high temperatures slow down the rate of anthocyanin biosynthesis in red grapes [43]. Thus, based on these previous studies, it is likely that both of these processes—i.e., pigment degradation and reduced anthocyanin biosynthetic rate—simultaneously take place in grape skins of Malbec and Pinot Noir under HT conditions, and may explain our results. Whether genetic or physiological variations in these mechanisms are related to the different response observed in Merlot, which maintained unaffected total anthocyanin levels under HT, is unclear and deserves further work.

A “thermal decoupling” phenomenon, consisting of a delayed anthocyanin accumulation (but not of sugar content) in berries ripened under HT, has been described in Cabernet Franc and Shiraz [44,45]. In the present study, pH and sugar content in berries of Malbec, Merlot, and Pinot Noir remained unaffected over the entire experimental period, suggesting that the thermal uncoupling phenomenon reported previously does not occur in these cultivars. Recently, it has been suggested that this decoupling mechanism is cultivar dependent [46], and it can also vary among different clones within a cultivar [47].

We found that the proportion of acylated anthocyanins increased in Malbec and Merlot berries under HT conditions (Figure 3). Similar results were reported for Cabernet Sauvignon [37], Sangiovese fruiting cuttings under different temperature regimes [39], and Merlot vines grown in chambers with increased temperature by air flow [40]. It has been previously demonstrated that acylation increases the chemical stability of anthocyanins [48]. Due to their antioxidant properties, these secondary metabolites are used by the plant to attenuate stressful situations, ameliorating cellular damage [48]. In the present work, the relative increase in AA observed in Malbec and Merlot berries under HT was mainly due to an increase in the coumarylated rather than acetylated forms of anthocyanins (Appendix A). The relative increase in AA is a desired characteristic in grapes destined for aged wines. The absence of AA in Pinot Noir was expected and corresponds to its typical anthocyanin profile. This cultivar presents a deleterious mutation in the Vvi3AT gene, which encodes a BAHD acyltransferase enzyme, necessary for anthocyanin acylation [36].

In the present study, resveratrol content was determined in berries of the three grape cultivars, but no significant differences were found between the HT and C treatments in both years (Appendix A). Jeandet et al. [26] proposed that chalcone synthase (EC 2.3.1.74) and stilbene (resveratrol) synthase (EC 2.3.1), two key enzymes of the flavonoid and stilbenoid biosynthetic pathways, compete for precursors for the production of anthocyanins and resveratrol, respectively, suggesting a counterbalanced equilibrium between the content of these two compounds (i.e., when anthocyanins are produced and their concentration increases, resveratrol content decreases; and vice versa). In apparent contradiction to the latter equilibrium, our results in Malbec and Pinot Noir showed that resveratrol content was unaffected by increased temperature while anthocyanin content was significantly reduced in the HT treatment (relative to Control plants). These results suggest that either: (1) anthocyanin and resveratrol production utilize independent sources of precursors for their biosynthesis; or (2) high temperature promotes mainly—or exclusively—degradation of already formed anthocyanins, but does not affect the biosynthesis of anthocyanin precursors, thereby reducing only pigment levels; or (3) HT promotes both anthocyanin degradation and reduced rate of anthocyanin biosynthesis, but the latter occurs by influencing some component(s) downstream from the pathway after divergence of the stilbenoid and flavonoid pathways. However, previous studies of Jeandet et al. [26] and Hrazdina et al. [49] provided supporting evidence, at the enzymatic and molecular levels, for the proposed negative relationship between resveratrol biosynthesis and anthocyanin accumulation in berry skins of different Vitis species. Additionally, in Sangiovese berries under HT conditions, anthocyanin biosynthesis was suppressed at both transcriptional and enzymatic levels, and peroxidase activity was increased, suggesting that both mechanisms—i.e., increased pigment degradation and reduced anthocyanin biosynthesis—are involved and account for the reduced pigment content in this cultivar under HT [42]. Altogether, these previous studies suggest that our first two hypotheses are less probable, despite the fact that they used different genotypes, and therefore genotype-based differences cannot be ruled out.

The fact that, in 2018, Merlot berries under HT conditions significantly increased the proportion of acylated anthocyanins throughout the entire ripening period while maintaining an unaffected total anthocyanin concentration (Figure 2F and Figure 3D) suggests that anthocyanin acylation could have increased pigment stability, thereby buffering HT stress consequences by ameliorating anthocyanin degradation. Such phenotypic plasticity, as suggested by the observed shift in anthocyanin profiles toward a higher proportion of more stable acylated pigments, may represent an adaptive response of Merlot berries to cope with HT conditions.

In Malbec berries under HT, although a significant reduction in total anthocyanin content was observed at veraison, this effect was overturned as ripening of the berries progressed, showing comparable (i.e., not statistically different) total pigment content between HT and C at half ripening and harvest (Figure 2B), concomitantly with significant increases in AA at these phenological stages (Figure 3B). Lastly, Pinot Noir berries under high temperature revealed unaffected anthocyanin concentrations at veraison and half ripeness, as compared to Control conditions, but at harvest time the thermal stress was accompanied by a significant reduction in total pigment content (Figure 2C,D). Based on these results and considering previous studies reporting that high temperatures activate anthocyanin acylation and, thereby, pigment stability in some cultivars (e.g., Cabernet Sauvignon [37], Malbec and Bonarda [43], Sangiovese [39], and Merlot [16,40]), we speculate that the lack of a functional acyltransferase enzyme that catalyzes anthocyanin acylation in Pinot Noir [36] causes this cultivar to be devoid of such stress-response mechanism. Altogether, the results of this study suggest high plasticity in response to climate change for Merlot, moderate plasticity for Malbec, and low plasticity for Pinot Noir.

## 4. Materials and Methods

### 4.1. Plant Materials, Temperature Conditions, and Experimental Design

Experiments were performed in a commercial vineyard located at Agrelo, Luján de Cuyo, Mendoza, Argentina (33°4′0.507″ S, 68°53′23.064″ W). Vines of cv. Malbec, Merlot and Pinot Noir were grown on a high vertical shoot position (VSP) trellis system with hail mesh, using furrow irrigation and conventional tillage management, in loam-sandy soil. In order to increase mean diurnal temperature on selected plants, transparent 100 microns nylons were placed from the ground to clusters level, and held with clamps to the hail mesh, on both sides of the plant rows, as depicted in Appendix A. These vines (20 plants) constituted the high temperature (HT) treatment. Control plants (C) (N = 20) were grown in the same way as the HT plants but lacked the nylon structure. Four replicates of 5 plants each were used for each temperature condition (HT and C). The temperature was monitored with iButton DS1921G 1 Wire^®^ Thermochron^®^ (Maxim Integrated, CA, USA) thermocouples placed next to the plant at cluster level.

Berries of Malbec were sampled in two seasons at 50% veraison (24 January 2017 and 30 January 2018), half ripeness (14 February 2017 and 6 February 2018), and harvest (17 March 2017 and 13 March 2018). Similarly, berries of Merlot were sampled at 50% veraison (24 January 2017 and 30 January 2018), half ripeness (14 February 2017 and 6 February 2018), and harvest (17 March 2017 and 20 February 2018); and berries of Pinot Noir were sampled at 50% veraison (24 January 2017 and 23 January 2018), half ripeness (31 January 2017 and 30 January 2018), and harvest (14 February 2017 and 13 February 2018). 

Berries were cut from the plant, immediately frozen with liquid nitrogen, and stored at −80 °C in the laboratory. Soluble solids content (°Brix) (Master refractometer, Atago, Japan) and pH (TPX-III, Altronix, Taiwan) for 50% veraison, half ripeness, and harvest were determined in the fruit juice of berry samples obtained from the same cluster used for biochemical analysis.

### 4.2. Phenolics Extraction and HPLC Analysis of Anthocyanins and Resveratrol

Grape berry skins were ground to powder in liquid nitrogen and their phenolic compounds were extracted according to Revilla et al. [50], with minor modifications. Briefly, the extracts were obtained in darkness by macerating 150 mg of skin powder with 1850 µL of acidified methanol, using a MeOH:HCl ratio of 99:1 (*v*/*v*) (HCl 10 N) (Sintorgan, Buenos Aires, Argentina), for 24 h. The extracts were then centrifuged at 4 °C and 14,000 rpm (Z 326K, Hermle Labortechnik GmbH, Wehingen, Germany) and the supernatants were recovered and stored at −20 °C, whereas the precipitates (pellets) were used for a second phenolic extraction with 1850 µL of solvent during 24 h in darkness. Equal parts of each supernatant were combined and filtered through a 45 µm pore cellulose acetate membrane (Sartorius, Göttingen, Germany), and analyzed by High Performance Liquid Chromatography with Diode-Array Detection (HPLC-DAD) (SPD-M10 AVP, Shimadzu Scientific Instruments, Maryland, MD, USA). Anthocyanins profile analysis was performed according to the protocol of the International Compendium of Analysis Methods of Musts and Wines [51]. HPLC analyses were performed using a Lichrosorb reverse-phase column (RP-18, 250 mm × 4.6 mm × 5 µm, Merck, Darmstadt, Germany). Anthocyanins were quantified at 520 nm using a calibration curve from a commercial standard of malvidin-3-glucoside chloride (Sigma, St. Louis, MO, USA). Anthocyanin concentration was expressed as mg per g of berry skin fresh weight (mg·g^−^^1^ FW).

Trans-resveratrol analysis was performed according to the protocol of the Argentine National Institute of Viticulture (INV). HPLC analyses were performed using a Kromasil reverse-phase (RP-18, 8 μm × 250 mm × 4.6 mm). The mobile phases were A: 100% acetonitrile (Merck, Darmstadt, Germany); and B: 3.4 mM formic acid (Sintorgan, Buenos Aires, Argentina). Elution was obtained with the following phase A gradient: 0% at 0 min, 20% at 3 min, 31% at 7 min, 32% at 11 min, 80% at 15 min, 100% at 16 min, 20% at 20 min. Flow rate was 1 mL/min with a sample injection of 20 µL. Absorbance was measured at 320 nm. Trans-resveratrol was quantified at 320 nm using a calibration curve from a commercial standard of 3,4′,5-Trihydroxy-trans-stilbene, resveratrol (Sigma, St. Louis, MO, USA). Resveratrol concentration was expressed as µg·g^−^^1^ of berry skin fresh weight (FW).

### 4.3. Statistical Analysis

The experimental design was a Latin Square with four replicates in each case. Differences in anthocyanin content were analyzed using a general linear and mixed model, mean comparisons were performed by LSD and Fisher tests, considering *p* values ≤ 0.05 significant. Differences in anthocyanin composition were analyzed by means comparisons, DGC test for *p* < 0.05 adjusting a general linear and mixed model. In both cases, adjustments were made for heteroscedasticity. The analysis was undertaken employing the software InfoStat, v2020p, Córdoba, Argentina.

## 5. Conclusions

This study performed—for the first time—side-by-side comparative analysis on the effect of heat stress associated with a realistic climate change scenario on anthocyanin concentration and composition, and pH, solids, and resveratrol content in berries of three major red grape cultivars, in a two-year replicated trial under typical field-grown conditions in one of the main viticulture regions of the world. Our results revealed cultivar-specific responses to HT conditions, with Malbec and Pinot Noir exhibiting significant decreases in total anthocyanin concentration at veraison and harvest, respectively, while Merlot maintained pigment content unaffected under HT at all phenological stages. The decrease in total pigment content in Malbec and Pinot Noir was associated with reduced levels of delphinidin, cyanidin, petunidin, peonidin, and malvidin glycosides, and with increases in the relative content (%) of acylated anthocyanins. These data suggest pigment acylation as a possible stress-response mechanism for attenuating HT negative effects. Altogether, these results suggest high plasticity in response to climate change for color attributes in Merlot, moderate plasticity for Malbec, and low plasticity for Pinot Noir. The other wine quality variables analyzed (pH, resveratrol, and solids content) were not affected by HT. Our data suggest that further investigation on individual and relevant wine-producing cultivars, concerning their behavior and plasticity under climate change conditions, will be necessary for future sustainable viticulture practices.

## Figures and Tables

**Figure 1 plants-11-00926-f001:**
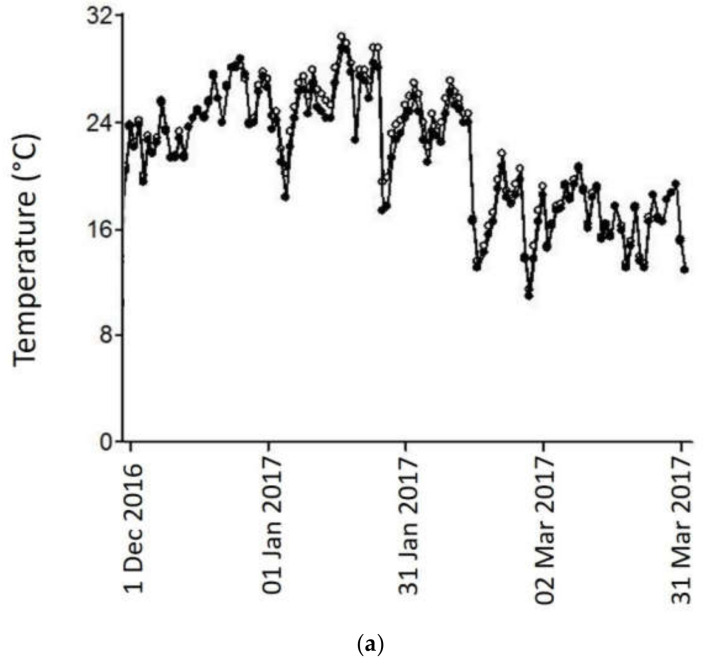
Daily mean diurnal temperatures (°C) for the high temperature (HT) and Control (C) treatments during the experimental period in seasons 2017 (**a**) and 2018 (**b**). Error bars represent standard errors for three replicates.

**Figure 2 plants-11-00926-f002:**
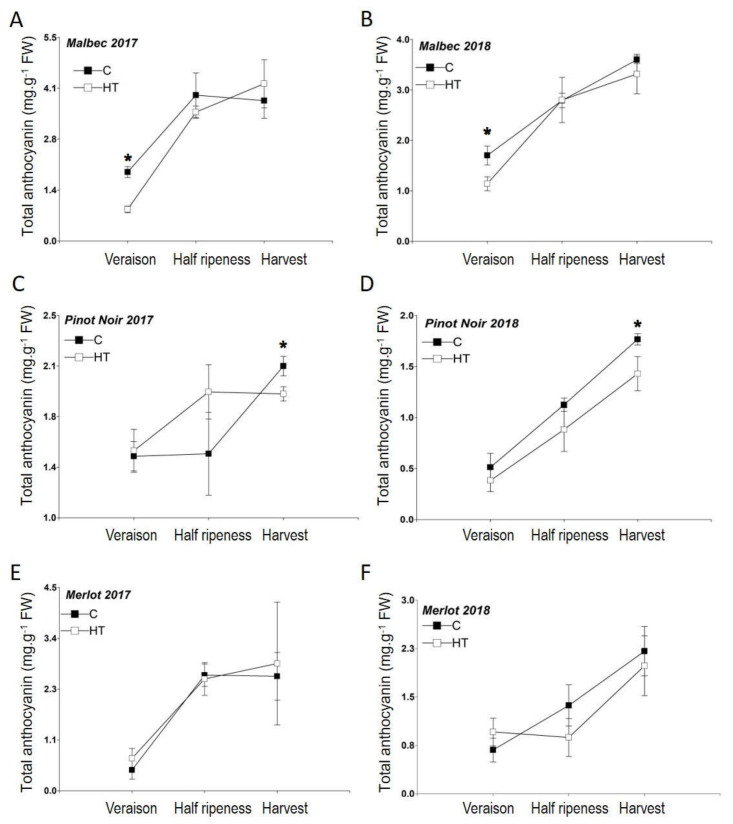
Total anthocyanin concentration in berries of Malbec (**A**,**B**), Pinot Noir (**C**,**D**), and Merlot (**E**,**F**) plants grown under increased temperature (HT) and normal conditions (**C**) in 2017 and 2018. Error bars represent standard errors from four replicates. Asterisks indicate significantly different (*p* < 0.05) anthocyanin concentration between HT and C treatments for a given phenological stage, according to means comparison analysis by LSD test.

**Figure 3 plants-11-00926-f003:**
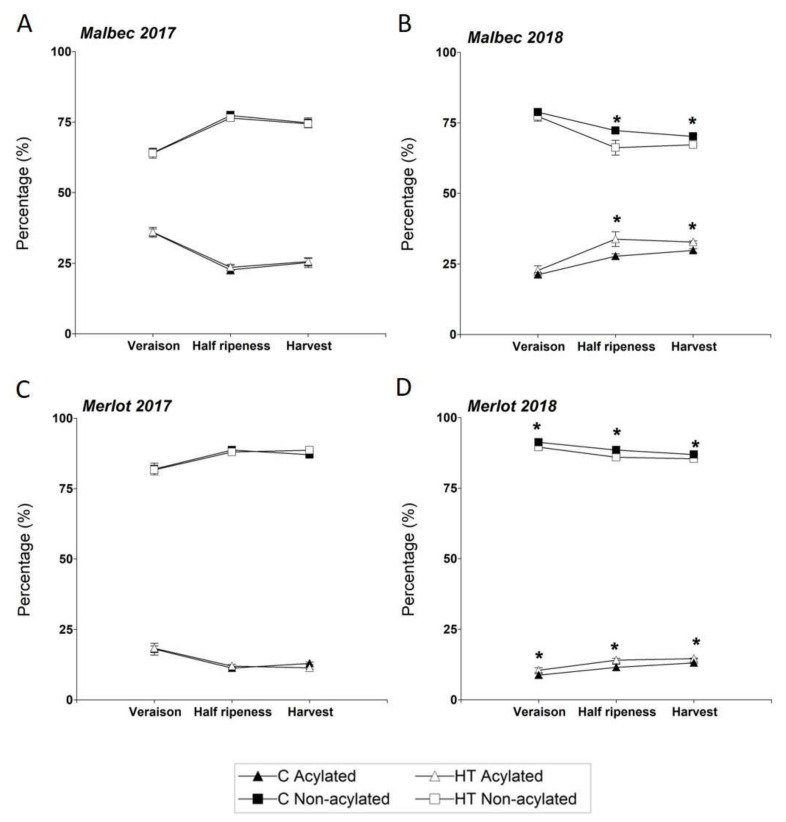
Relative content (%) of acylated and non-acylated anthocyanins in berries of Malbec (**A**,**B**) and Merlot (**C**,**D**) grown under high (HT) and control temperature (**C**) conditions in 2017 and 2018. Error bars represent standard errors from four replicates. * indicate statistical difference (*p* < 0.05) between treatments at a given phenological stage (LSD test).

**Table 1 plants-11-00926-t001:** Accumulated degree-days (DDs) and pH and soluble solids content (°Brix) in berries of Malbec, Merlot, and Pinot Noir plants grown under increased temperature (High temperature, HT) and natural conditions (Control, C), for three phenological stages (veraison, half-ripeness, harvest) in seasons 2017 and 2018. Values are means of four replicates. ns, no statistical difference (*p* < 0.05) between HT and C, according to Kruskal Wallis test.

Year	Treatment		Malbec						Merlot						Pinot Noir					
			Veraison		Half Ripeness		Harvest		Veraison		Half Ripeness		Harvest		Veraison		Half Ripeness		Harvest	
		DDs	SS ^§^	pH	SS	pH	SS	pH	SS	pH	SS	pH	SS	pH	SS	pH	S	pH	SS	pH
2017	HT	541	11.9	3.2	18.6	3.4	24.0	3.4	11.5	3.1	20.6	3.6	23.6	3.7	16.1	3.7	19.5	3.8	21.2	4.0
	C	482	11.5	3.1	18.0	3.3	23.8	3.3	12.5	3.1	19.8	3.5	24.0	3.6	16.9	3.6	18.1	3.7	21.6	3.8
			ns	ns	ns	ns	ns	ns	ns	ns	ns	ns	ns	ns	ns	ns	ns	ns	ns	ns
2018	HT	706	14.7	3.0	17.6	3.3	24.9	3.9	13.8	3.2	18.0	3.5	22.3	3.8	12.8	3.1	16.4	3.6	21.8	4.0
	C	612	15.3	3.0	17.0	3.3	24.1	3.8	13.6	3.1	18.1	3.4	23.5	3.8	13.1	3.2	15.7	3.4	20.3	3.8
			ns	ns	ns	ns	ns	ns	ns	ns	ns	ns	ns	ns	ns	ns	ns	ns	ns	ns

^§^ Soluble solids (SS) are expressed in °Brix. ns. not significantly different between HT and C.

**Table 2 plants-11-00926-t002:** Grape anthocyanin pigments with approximate HPLC retention times.

Compound	Abbreviation	RT
delphinidin-3-glucoside	Df	12.87
cyanidin-3-glucoside	Cn	15.33
petunidin-3-glucoside	Pt	17.11
peonidin-3-glucoside	Po	19.57
malvidin-3-glucoside	Mv	20.85
peonidin-3-O-acetylglucoside	PoAc	28.81
malvidin-3-O-acetyl-glucoside	MvAc	29.30
peonidin-3-O-coumaroyl-glucoside	PoCu	34.56
malvidin-3-O-coumaroyl-glucoside	MvCu	35.33

RT is retention time (min) for the chromatographic procedure described by the OIV [25].

**Table 3 plants-11-00926-t003:** Relative content (%) of nine anthocyanin pigments in berries of Malbec, Merlot, and Pinot Noir plants grown under high (HT) and control temperature (C) conditions during the fruit ripening process.

Anthocyanin Pigment ^§^/Year	MALBEC					
	Veraison		Half Ripeness		Harvest	
	HT	C	HT	C	HT	C
**Df**						
2017	4.3 ± 0.3	4.5 ± 0.2	**4.5 ± 0.3**	**6.1 ± 0.6 ***	4.3 ± 0.3	4.8 ± 0.4
2018	6.7 ± 0.6	7.5 ± 0.4	4.7 ± 0.4	5.4 ± 0.3	4.3 ± 0.1	4.5 ± 0.3
**Cn**						
2017	1.0 ± 0.1	1.1 ± 0.1	1.2 ± 0.1	1.0 ± 0.2	1.1 ± 0.1	1.0 ± 0.1
2018	**0.8 ± 0.1**	**1.1 ± 0.1 ***	0.4 ± 0.1	0.4 ± 0.1	0.1 ± 0.01	0.1 ± 0.02
**Pt**						
2017	8.4 ± 0.3	8.5 ± 0.3	9.5 ± 0.4	10.7 ± 0.7	8.2 ± 0.3	9.0 ± 0.6
2018	11.1 ± 0.7	11.9 ± 0.3	8.4 ± 0.6	9.4 ± 0.2	8.0 ± 0.1	8.3 ± 0.3
**Po**						
2017	3.7 ± 0.4	3.6 ± 0.2	**6.3 ± 0.6 ***	**4.2 ± 0.1**	**6.7 ± 0.5 ***	**4.7 ± 0.4**
2018	4.4 ± 0.1	4.8 ± 0.3	2.0 ± 0.1	2.7 ± 0.3	0.3 ± 0.03	1.7 ± 0.8
**Mv**						
2017	46.6 ± 0.8	46.5 ± 0.7	54.9 ± 1.1	55.2 ± 0.4	54.0 ± 0.9	55.3 ± 1.2
2018	54.3 ± 0.5	53.5 ± 0.8	**50.7 ± 1.4**	**54.4 ± 0.9 ***	**54.6 ± 0.3**	**55.6 ± 0.40 ***
**PoAc**						
2017	0.3 ± 0.02	0.3 ± 0.02	**0.2 ± 0.02 ****	**0.2 ± 0.02**	**0.1 ± 0.01 ***	**0.1 ± 0.01**
2018	0.2 ± 0.02	0.2 ± 0.02	0.2 ± 0.04	0.2 ± 0.02	**0.1 ± 0.01 ***	**0.1 ± 0.00**
**MvAc**						
2017	**5.0 ± 0.2**	**5.4 ± 0.03 ***	3.0 ± 0.2	3.1 ± 0.4	1.4 ± 0.08	1.4 ± 0.06
2018	0.8 ± 0.02	1.1 ± 0.3	**0.6 ± 0.1 ***	**0.3 ± 0.1**	0.1 ± 0.00	0.2 ± 0.01
**PoCu**						
2017	**2.8 ± 0.1 ***	**2.5 ± 0.02**	**1.8 ± 0.2 ***	**1.2 ± 0.04**	**2.2 ± 0.3 ***	**1.5 ± 0.2**
2018	2.0 ± 0.1	1.9 ± 0.2	1.5 ± 0.1	1.3 ± 0.01	**1.6 ± 0.1 ***	**1.3 ± 0.1**
**MvCu**						
2017	27.9 ± 1.7	27.7 ± 1.3	18.4 ± 1.1	18.2 ± 1.6	21.9 ± 1.0	22.2 ± 1.6
2018	19.8 ± 1.6	18.1 ± 0.8	**31.5 ± 2.6 ***	**26.0 ± 0.9**	**30.9 ± 0.4 ***	**28.2 ± 0.7**
**Total AA**						
2017	36.0	35.9	23.6	22.7	25.6	25.2
2018	22.7	21.2	**33.8 ***	**27.7**	**32.7 ***	**29.8**
**Df**						
2017	5.7 ± 0.4	5.5 ± 0.5	**6.3 ± 0.2**	**8.5 ± 0.5 ***	**7.8 ± 0.8 ****	**7.0 ± 0.5**
2018	**8.8 ± 1.0**	**13.1 ± 1.3 ***	7.0 ± 0.8	9.7 ± 1.3	**6.6 ± 0.4**	**8.1 ± 0.3 ***
**Cn**						
2017	2.8 ± 0.3	2.9 ± 0.4	3.5 ± 0.4	3.7 ± 0.1	**5.7 ± 0.8 ***	**3.3 ± 0.4**
2018	**2.2 ± 0.6**	**3.8 ± 1.0 ***	**1.1 ± 0.2**	**2.0 ± 0.4 ***	**1.2 ± 0.1**	**2.2 ± 0.3 ***
**Pt**						
2017	9.1 ± 0.4	9.3 ± 0.4	**11.0 ± 0.2**	**12.7 ± 0.5 ***	10.8 ± 0.6	10.7 ± 0.3
2018	**12.5 ± 0.8**	**13.9 ± 0.8 ****	10.5 ± 0.6	12.3 ± 0.9	**9.5 ± 0.3**	**10.7 ± 0.3 ***
**Po**						
2017	10.1 ± 0.8	11.6 ± 0.6	10.8 ± 1.2	9.7 ± 0.6	**13.9 ± 0.8 ***	**10.2 ± 0.9**
2018	**6.8 ± 1.1**	**8.9 ± 1.3 ***	**4.6 ± 0.5**	**7.0 ± 0.8 ***	**4.9 ± 0.4**	**7.4 ± 0.8 ***
**Mv**						
2017	53.9 ± 0.8	52.8 ± 2.3	**56.5 ± 1.0 ***	**54.2 ± 0.8**	**50.4 ± 1.2**	**55.9 ± 1.3 ***
2018	**59.3 ± 2.5 ****	**51.6 ± 3.2**	**62.8 ± 1.8 ***	**57.6 ± 2.2**	**63.2 ± 0.9 ****	**58.5 ± 1.2**
**PoAc**						
2017	0.9 ± 0.1	1.0 ± 0.1	0.5 ± 0.04	0.5 ± 0.01	0.2 ± 0.03	0.2 ± 0.03
2018	**0.1 ± 0.01 ****	**0.1 ± 0.01**	**0.2 ± 0.02 ***	**0.1 ± 0.01**	**0.1 ± 0.00 ****	**0.07 ± 0.00**
**MvAc**						
2017	7.2 ± 0.2	6.7 ± 0.5	3.7 ± 0.5	3.9 ± 0.2	**1.4 ± 0.2**	**1.9 ± 0.1 ***
2018	**0.3 ± 0.05 ***	**0.1 ± 0.02**	0.2 ± 0.04	0.1 ± 0.01	**0.4 ± 0.00 ***	**0.3 ± 0.05**
**PoCu**						
2017	1.4 ± 0.1	1.3 ± 0.3	**1.2 ± 0.1 ***	**1.0 ± 0.1**	**2.1 ± 0.1 ***	**1.7 ± 0.1**
2018	**1.2 ± 0.1**	**1.4 ± 0.1 ****	1.3 ± 0.1	1.3 ± 0.1	**1.3 ± 0.1**	**1.8 ± 0.2 ***
**MvCu**						
2017	8.9 ± 0.6	9.0 ± 2.1	**6.6 ± 0.1 ***	**5.9 ± 0.2**	**7.7 ± 0.2**	**9.1 ± 0.4 ***
2018	**8.9 ± 1.0 ***	**7.1 ± 0.9**	**12.4 ± 0.8 ***	**10.0 ± 1.1**	**12.7 ± 0.2***	**10.9 ± 0.4**
**Total AA**						
2017	18.3	18.0	12.0	11.3	11.4	13.0
2018	**10.5 ***	**8.7**	**14.0 ***	**11.5**	**14.6 ***	**13.1**
**Df**						
2017	**0.6 ± 0.1 ***	**0.4 ± 0.1**	0.9 ± 0.02	0.8 ± 0.2	**1.3 ± 0.1 ***	**1.1 ± 0.03**
2018	**1.5 ± 0.4**	**3.0 ± 0.3 ***	**2.0 ± 0.2**	**3.9 ± 0.1 ****	3.2 ± 0.3	3.6 ± 0.3
**Cn**						
2017	0.3 ± 0.05	0.2 ± 0.04	0.5 ± 0.01	0.5 ± 0.1	**1.0 ± 0.1 ***	**0.8 ± 0.1**
2018	0.9 ± 0.2	1.0 ± 0.1	**1.3 ± 0.1 ***	**1.1 ± 0.04**	**1.8 ± 0.3**	**2.3 ± 0.01 ***
**Pt**						
2017	**4.0 ± 0.4 ***	**3.0 ± 0.6**	4.4 ± 0.1	3.8 ± 0.6	**4.8 ± 0.1 ***	**4.4 ± 0.1**
2018	**4.4 ± 0.7**	**6.9 ± 0.5 ***	**5.2 ± 0.2**	**7.6 ± 0.2 ****	6.1 ± 0.4	6.2 ± 0.4
**Po**						
2017	9.2 ± 0.8	10.8 ± 0.3	12.2 ± 0.9	13.5 ± 1.3	17.6 ± 0.6	16.2 ± 1.5
2018	16.1 ± 3.0	12.1 ± 0.3	18.6 ± 1.7	15.6 ± 1.0	21.6 ± 3.1	23.7 ± 1.0
**Mv**						
2017	85.9 ± 0.5	85.7 ± 0.8	82.0 ± 0.8	81.4 ± 1.8	75.4 ± 0.6	77.6 ± 1.7
2018	77.3 ± 2.8	77.2 ± 0.9	73.0 ± 1.4	72.0 ± 1.0	67.3 ± 3.0	64.3 ± 0.6
**PoAc**						
2017	ND	ND	ND	ND	ND	ND
2018	ND	ND	ND	ND	ND	ND
**MvAc**						
2017	ND	ND	ND	ND	ND	ND
2018	ND	ND	ND	ND	ND	ND
**PoCu**						
2017	ND	ND	ND	ND	ND	ND
2018	ND	ND	ND	ND	ND	ND
**MvCu**						
2017	ND	ND	ND	ND	ND	ND
2018	ND	ND	ND	ND	ND	ND
**Total AA**						
2017	0.0	0.0	0.0	0.0	0.0	0.0
2018	0.0	0.0	0.0	0.0	0.0	0.0

^§^ Abbreviated pigments names are as follows: Df. delphinidin-3-glucoside; Cn. cyanidin-3-glucoside; Pt. petunidin-3-glucoside; Po. peonidin-3-glucoside; Mv. malvidin-3-glucoside; PoAc. peonidin-3-O-acetylglucoside; MvAc. malvidin-3-O-acetyl-glucoside; PoCu. peonidin-3-O-coumaroyl-glucoside; MvCu. malvidin-3-O-coumaroyl-glucoside. Values are means ± standard errors of four replicates, expressed as percentage of the total anthocyanin content. Compounds significantly affected by the temperature treatments are denoted in bold letters. Asterisks indicate significant difference between HT and C treatments at *p* ≤ 0.05 (*), *p* ≤ 0.01 (**), for a given year, phenological stage, and cultivar (DGC test).

## Data Availability

All data have been included in the paper.

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
