# Peer review of "High Temperature Alters Anthocyanin Concentration and Composition in Grape Berries of Malbec, Merlot, and Pinot Noir in a Cultivar-Dependent Manner"

_plants, 2022, doi:10.3390/plants11070926_

Round 1

Reviewer 1 Report

The research is focused on High temperature alters anthocyanin concentration and composition in grape berries of Malbec, Merlot, and Pinot Noir in a cultivar-dependent manner. I appreciate the Results part, which is very detailed. Great attention I suggest for the shape of the manuscript, which is equal important as the content. In this regard I suggest checking again the Instructions for authors you can find at the following link https://www.mdpi.com/journal/plants/instructions 

Please see bellow my suggestions, in order to facilitate the improvement of the manuscript: 

Please separate the keywords with semicolon, not with comma. 

L62. Please complete after ref. [7-10], their specific chemical structure imprinting them characteristic antioxidant proprieties [Glevitzky I., et al. Statistical Analysis of the Relationship Between Antioxidant Activity and the Structure of Flavonoid Compounds. Rev. Chim. 2019, 70(9), 3103-3107. https://doi.org/10.37358/RC.19.9.7497]

Furthermore, describing the climate it must be also reminded the soil quality/type and the management of it as being of the utmost importance in the quality of the grapes/wine. Together, the climate and soil composition are defining. Please develop this idea according to Bungau et al. Expatiating the impact of anthropogenic aspects and climatic factors on long term soil monitoring and management. Environ Sci. Pollut. Res. 2021, 202, 30528-30550. https://doi.org/10.1007/s11356-021-14127-7 ; Samuel A.D.,et al. Enzymological and physicochemical evaluation of the effects of soil management practices. Rev. Chim. 2017, 68(10), 2243-2247. https://doi.org/10.37358/RC.17.10.5864 ; 

Please make the actual aim (L91-94) of study relevant, in a SEPARATE, LAST paragraph of Introduction to increase its visibility. Responding to the following questions would be helpful: What makes special this study? Which is its novelty character or its special aspects? Why have the author chosen this topic? What differentiate this paper from others in the same/similar topic? Actual text is not so relevant, not being enough to justify the relevance and necessity of publishing your study.

Table 3. It is not clear why some numerical values are bolded. Please explain under the table or unbold them.

L336. Beginning the paragraph with "In conclusion..." it seems that there are the Conclusions of the paper. Please reshape as to not be confusing, existing a separate section of 5. Conclusions where you can move the paragraph L336-351

Section 4. Please check and provide the Model, Producer/manufacturer, City and Country for each apparatus used in the research, and the Producer and Country for each reagent/chemical used. 

4.3. Please provide the computer programs used and their variants.

5. Conclusions must be completed with paragraph 336-351. 

References should be written in the MDPI style, providing all data, according to the Instructions for authors - please check them.

Reviewer 2 Report

The authors propose a manuscript titled “High temperature alters anthocyanin concentration and composition in grape berries of Malbec, Merlot, and Pinot Noir in a cultivar-dependent manner”.

The article is well structured and argued. In particular, this study takes into consideration a on a crucial topic about that the climate is determinant for grapevine geographical distribution, and consequently on berry attributes and finally on wine quality. Due to a increase in mean diurnal temperature in Argentina country the authors hypothesize that this climate parameter will affect color intensity and other quality attributes of red grapes and wines. In particular the sutdy proposed highligths the effect of high temperature on anthocyanin concentration and composition, pH, and resveratrol and solids content in berries of three major wine-producing varieties (Malbec, Merlot and Pinot Noir) during fruit ripening in two different seasons. The results suggest that the pigment acylation as a possible stress-response mechanism for attenuating negative effects of high temperature..

I appreciate the work which is written correctly for agronomic point of view. The figures and the tables are clear for the reader. However, I have highlighted and suggested some crucial points to be further explored, which I believe the authors will have no problem to satisfying.

  1. Introduction

I suggest to add a brief period on world distribution of the three grapewine varieties here considered in order to give a complete picture at the readers.

Please complete as I suggested (in bold). Sometimes several statement, correctly argued are without reference, why?

  • Lines 41-42. “Grapevine cultivation in different biogeographical re gions demonstrates the great plasticity of this crop plant [choose a reference]”;
  • Lines 47-48 “…in addition to implementing cultivar-specific agricultural practices of the vineyard [5], better if with extensive type in order to preserve the characteristics of the agroecosystem and environments parameters [Perrino and Calabrese 2018];
  • Lines 49-50. “The climate in western central Argentina allows the production of high-quality grapes and wines [choose a reference]”;
  • Lines 52-56. “The climatological characteristics of this region are 52 optimal for producing high quality grapes and wines [choose a reference]. Malbec, Merlot and Pinot Noir wines from Mendoza are internationally recognized for their high-quality organoleptic characteristics [choose a reference]. Among them, color intensity, hue, and stability are the main color quality traits in red wines. These attributes are strongly conditioned by the presence, concentration, and composition of anthocyanins in the grape berry skin, along with other constituents that influence color development and stability in grapes and wines (e.g., other non- anthocyanin phenolic compounds) [choose a reference].
  • Lines 80-82. The authors declare correctly a general and crucial concept, but valid also for each wild plants and not only for crop, inlcuding grapewine. I suggest to add a refernce valid for wild plants “Phenotypic plasticity (or adaptability), defined as the ability of an organism (wild and cultivated) to change its phenotype in response to different environments, can help mitigate negative effects due to environmental perturbations [20, Perrino and Wagensommer 2022];
  • Lines 89-91. In addition, this information would contribute to mapping adequate regions for growing different cultivars under the predicted climate modifications [choose a reference].

Reference to be added:

  • Perrino, E.V.; Calabrese G., Vascular flora of vineyards in the DOC area “Gioia del Colle” (Apulia, Southern Italy): preliminary data. Natura Croatica, 2018, 27, 41-55. https://doi.org/20302/NC.2018.27.3
  • Perrino, E.V.; Wagensommer, R.P. Crop Wild Relatives (CWRs) Threatened and Endemic to Italy: Urgent Actions for Protection and Use. Biology 2022, 11, 193. https://doi.org/10.3390/biology11020193
  1. Results

Well done, te tables and figures are clear. Few suggestions.

  • Only in table 2 please check value of RT;
  • Tables 3, page 5 and 7. pigments instead pigment§
  1. Discussion

Well done. Few suggestions.

  • Line 217. The three grapewine varieties studied;
  • Line 264. phenomenon consisting instead phenomenon -consisting
  1. Materials and Methods

Please give a geographical position, better if with map and geographic coordinates, of vineyard located at Agrelo, Luján de Cuyo and Mendoza.

  • Line 408. the software used Statistical analysisis not clear
  1. Conclusion

Please spend two word on future prospectives research in the topic disccussed in the manuscript.

References

Please follow the guidelines of the journal and give DOI when is available.

Round 2

Reviewer 1 Report

The Authors responded to my requests.

Reviewer 2 Report

Dear authors, I appreciate the effort made to update the manuscript, following my suggestions.  I believe this version is publishable

Reviewer